# Comparison of the Phytochemical Variation of Non-Volatile Metabolites within Mother Tinctures of *Arnica montana* Prepared from Fresh and Dried Whole Plant Using UHPLC-HRMS Fingerprinting and Chemometric Analysis

**DOI:** 10.3390/molecules27092737

**Published:** 2022-04-24

**Authors:** Simon Duthen, Alice Gadéa, Pascal Trempat, Naoual Boujedaini, Nicolas Fabre

**Affiliations:** 1UMR 152 PharmaDev, Université de Toulouse, IRD, UPS, 31062 Toulouse, France; simon.duthen@gmail.com (S.D.); alice.gadea@univ-tlse3.fr (A.G.); 2Boiron Lab, 69510 Messimy, France; pascal.trempat@boiron.fr (P.T.); naoual.boujedaini@gmail.com (N.B.)

**Keywords:** *Arnica montana*, metabolomics, mother tinctures, phytochemical differences, dried vs. fresh materials

## Abstract

*Arnica montana* L. has been recognized for centuries as an herbal remedy to treat wounds and promote healing. It also has a long tradition of use in homeopathy. Depending on its medicinal utilization, standardization regulations allow different manufacturing processes, implying different raw materials, such as the whole arnica plant in its fresh or dried state. In this study, an untargeted metabolomics approach with UHPLC-HRMS/MS was used to cross-compare the phytochemical composition of mother tinctures of *A. montana* that were prepared from either fresh whole plant (fMT) matter or from oven-dried whole plant (dMT) matter. The multivariate data analysis showed significant differences between fMT and dMT. The dereplication of the HRMS and MS/MS spectra of the more discriminant compounds led to annotated quinic acid, dicaffeoyl quinic acids, ethyl caffeate, thymol derivatives and dehydrophytosphingosine, which were increased in fMT, while Amadori rearrangement products (ARP) and methoxyoxaloyl-dicaffeoyl quinic acid esters were enhanced in dMT. Neither sesquiterpene lactones nor flavonoids were affected by the drying process. This is the first time that a sphingosine, ethyl caffeate and ARP are described in *A. montana*. Moreover, putative new natural products were detected as 10-hydroxy-8,9-epoxy-thymolisobutyrate and an oxidized proline fructose conjugate, for which isolation and full structure elucidation will be necessary to verify this finding.

## 1. Introduction

*Arnica montana* L. (Asteraceae) is an herbaceous perennial plant. It grows in the wild across Europe in meadows, pastures and heaths, mainly in the mountains, from south Norway and Latvia to south Portugal, the North Apennines and South Carpathians [1,2,3]. *Arnica* flowers have been known for centuries as an herbal remedy, originally being used internally in phytotherapy, and currently only applied externally on injuries, such as bruises, sprains, inflammation caused by insect bites, gingivitis, aphthous ulcers and the symptomatic treatment of rheumatic muscle and joint complaints [4,5,6,7]. *A. montana* also has a very long history with regard to its homeopathic utilization since it was originally described by Samuel Hahnemann, the inventor of this form of therapy [8].

The chemical composition of *A. montana* is well known; more than one hundred and fifty therapeutically active substances have been described in the different parts of the plant. As an overview, it contains sesquiterpene lactones of the pseudoguaianolide type, principally helenalin and 11α,13-dihydrohelenalin and their esters, with acetic, isobutyric, methacrylic, tiglic and other carboxylic acids. Other constituents include diterpenes, triterpenes (Arnidiol), flavonols and their glycosides and glucuronides, pyrrolizidine alkaloids (tussilagine and isotussilagine), polyacetylenes, chlorogenic acid derivatives (caffeoyl, cinnamoyl, feruloyl, fumaroyl, methoxyoxaloyl esters, coumarins, fatty acids, carotenoids and essential oil (for reviews, see [4,9]).

According to the European Pharmacopeia (monograph 1391), *A. montana* is defined in medicinal terms as the “whole or partially broken dried flower-heads of *Arnica montana* L.” [10]. According to the Herbal Medicinal Products Committee (HMPC) of the European Medicines Agency (EMA), herbal preparations for phytopharmaceuticals containing *A. montana* are a tincture of the dried flowers (drug extract ratio (DER) 1:10) extracted with EtOH at 70%, a tincture of the dried flowers (DER 1:10 or 1:5) extracted with EtOH at 60%, or a liquid extract of the fresh flowers (DER 1:20) obtained with EtOH at 50% [5]. It is worth mentioning that for the homeopathic preparations described in the European Pharmacopeia, the mother tincture of arnica (monograph 2371; method 1.1.10 referencing the French Pharmacopeia [11]) is obtained from the fresh whole flowering plant, extracted with EtOH at 45% (DER 1:10), and is corroborated by the homeopathic materia medica [12].

Over the last decade, metabolomics approaches have been developed that aim to identify and quantify all components within complex mixtures, whereby all of the acquired data can be leveraged to compare the phytochemical compositions between extracts [13,14]. However, the ability of metabolomics to encompass the full complexity of phytochemical blends greatly depends on the method used to capture the data. For instance, it has been shown that separative methods provide more accurate statistical models and higher identification rates compared to direct spectroscopic fingerprints [15].

Therefore, the aim of this work was to assess whether differences in chemical composition are detectable in homeopathic mother tinctures (MT) prepared from *A. montana* fresh whole plant (fMT) or dried whole plant (dMT) matter, using an LC-MS-based metabolomic approach without modification of the native molecules. For this, six sets of *A. montana* whole-plant matter were obtained from six different suppliers and divided into two parts, one of which was dried using an industrial process involving a multi-layer dryer heated to 65 °C. Twelve mother tinctures were prepared from raw materials and sampled in duplicate, leading to a total of 24 samples (set 1: fMT1, fMT2, dMT1, dMT2; set 2: fMT1, fMT2, …). All LC-MS fingerprints were then submitted to multivariate data analysis in order to assess molecular differences and to putatively identify the more discriminant compounds between fMT and dMT.

## 2. Results

### 2.1. Comparison of Dry Residues

As a preliminary step, the dry residues of MT obtained from fresh (fMT) and dried (dMT) whole plants were evaluated to confirm that the samples are quantitatively comparable for a metabolomic study. As shown in Appendix A, dry residues of fMT have an average of 2.597% (*m*/*v*) with a variation between samples ranging from 1.977 to 3.035% (*n* = 12). The mean of dry residues for dMT was 2.664%, ranging from 1.800 to 3.315% (*n* = 12). Student’s t-test of the means shows no significant difference.

### 2.2. Multivariate Data Analysis

The UHPLC-HRMS analysis of the 24 samples of fTM and dTM of *A. montana* and data processing using MSDial allowed the creation of a peak list of 924 features. Principal component analysis (PCA) was applied as exploratory data analysis to provide an unsupervised overview of the chromatogram fingerprints (Figure 1).

The representation of the PCA on the first two components reflects 60.5% of the total sample variance (2.2% for the principal component 1 (PC1) and 18.3% for principal component 2 (PC2). Several distinct clusters appear in the score plots of PC1 and PC2 and, as expected, the quality control samples, QCs, are positioned near the center of the PCA since they contain an aliquot of each MT. A reproducible response was observed as all independent replicates from the same group clustered together. Regarding the PC1, the main trend seems to be related to the production batches (different suppliers) of the mother tinctures. The distribution along the PC2 is clearly due to the state, whether fresh or dry, of the plant raw material used to prepare MTs.

### 2.3. Supervised Multivariate Analysis

According to the unsupervised PCA, showing that the state of plant raw material was a driving force involved in the separation of the various MTs, an orthogonal partial least squares discriminant analysis (OPLS-DA) model was carried out, filtering features according to the fresh and dried state of the raw material (Figure 2A). An R^2^Y of 0.988 and a Q^2^ of 0.965 demonstrate not only a good ability to discriminate between fMT and dMT but also a very good ability to predict the state of the raw material used to prepare a new MT. Thus, considering the small number of samples analyzed, the model can be considered robust. The part of the variance that is linked to the difference between the two classes corresponds to 17.7% (R^2^*p*(X) = 0.177). The OPLS-DA loading S-plot (Figure 2B) shows the covariance against the correlation of variables of the discriminating components. In this representation, the most significant metabolic features are distant from the center of the graph. Cut-off values for *p* (corr) > |0.6| and *p* > |0.05| were used to select metabolites that most influence the differences between f- and dMT. With the threshold values chosen in the OPLS-DA S-plot analysis, a total of 15 Rt-*m*/*z* couples were detected as being increased in one or other of the MT.

A second approach by OPLS-DA (data not shown) was carried out with a higher threshold, in order to only select on the chromatograms the most ionized compounds in negative and positive ion modes. The goal was to determine the role of the most intense compounds in LC-MS chromatograms that are able to discriminate between fMT and dMT. However, for the developed model, an R^2^(Y) of 0.745 and a Q^2^ of 0.43 with R^2^(Y) >> Q^2^ indicate that this supervised analysis has difficulty in discriminating between the fresh and dried raw materials. However, the application of a higher threshold led to 71 features (instead of 924) and highlighted two additional compounds at *m*/*z* 197 (negative ion) and 251 (positive ion) that play a role in sample discrimination. In the global analysis (model with 924 peaks), these two features are less influential and are located in the central part of the S-plot (Figure 2B). In spite of this finding, we thought that it would be interesting to add these 2 features to the previous 15 compounds, leading to 17 distinguishing markers between fMT and dMT that now need to be putatively identified. All distinguishing compounds are summarized in Table 1.

### 2.4. Dereplication of Discriminating Compounds

In order to annotate the discriminant compounds, each molecular ion was dereplicated “manually” (i.e., without any automated processing of the molecular formula with databases) with the help of high-resolution MS, MS/MS fragmentation, UV spectrum analysis, chemotaxonomy and the literature.

Looking at the enhanced products in the fMT (Table 1), it is clear that they belong to different chemical groups. This fact explains why the distinguishing molecular ions detected by the OPLS-DA belong to both polarities and are of the type [M+H]^+^ and [M-H]^−^ in terms of positive and negative ion ESI-MS detection, respectively. The “fold” column reports the intensity ratio of a molecular ion present in the fMT batches, relative to that same molecular ion present in the dMT batches. This ratio ranges from 1.3 for a thymol derivative to 3.1 for 4-hydroxy-8-sphingenine.

Concerning the annotation of quinic acid (QA), fumaric acid, 1,5-dicaffeoyl- and 3,5-dicaffeoyl quinic acids (DCQA), they have previously been described in *A. montana* [7,16,17] and their MS/MS fragmentations support the proposed structures. To distinguish the 6 (1,3-1,4-1,5-3,4-3,5 or 4,5) DCQA regioisomers identified in *A. montana*, Lin and Harnly [16] established a method based on their elution order under various LC conditions. Moreover, Clifford and colleagues [33,34] have developed solutions for discriminating between QA derivatives isomers based on the intensities of fragments detected by negative ion ESI-MS^n^. Thus, by combining the orders of elution and fragmentation patterns of the parent *m*/*z* 515 ions, the two DCQA isomers increased in fMT were unambiguously discriminated and annotated as 1,5- and 3,5-DCQA (Appendix A).

The peak at Rt 31.9 min was easily annotated to ethyl caffeate, according to its molecular formula, characteristic UV spectrum and fragmentation pattern (Appendix A). Caffeic acid ethyl ester has been described in several Asteraceae species, as reported by De Athayde and colleagues in a recent article [18], but has not been identified in a mother tincture prepared from *A. montana*.

The *m*/*z* 316 positive molecular ion, eluted at Rt 50.89 min and increased by more than 3 times in the fMT when compared to dMT, corresponds to a compound having the molecular formula C_18_H_37_O_3_N. The latter formula refers to 39 occurrences in the dictionary of natural products (DNP) database [35]. They all corresponded to trihydroxy amino C18 alkyl chains, also called sphingolipid long-chain bases and, more specifically, phytosphingosines when occurring in plants [19]. The proposed structure, 4-hydroxy-8-sphingenine, is supported by chemotaxonomic considerations since C18 di- or tri-OH sphingamines are common backbones of plants; when a double bond occurs in the structure, it is located at C-4 and/or C-8, with a predominance at C-8 for tri-OH derivatives [20,21]. The MS/MS spectrum depicted in Appendix A confirms the presence of 3-hydroxyl groups. The double bond can occur as an (*E*)- or (*Z*)-isomer, and the configurations at C-2, C-3 and C-4 are invariant (all trihydroxy sphingoid bases have a D-*ribo* configuration) [21].

The compound at Rt 37.07 min was putatively identified as a thymol derivative on the basis of two main arguments: its fragmentation pattern, illustrated in Figure 3, and the fact that it is the 10-deacetylated derivative of 10-acetoxy-8,9-epoxy-thymolisobutyrate, previously isolated in a tincture prepared from fresh flowerheads of *A. montana* [22]. It is worth noting that various thymol derivatives have been reported in the Asteraceae [23,24,25] and in *A. montana* roots [36,37], but the present proposed derivative has never been described in nature. Therefore, according to chemotaxonomic considerations (20 derivatives have previously been isolated in the genus *Eupatorium* [24,25]), the suggested structure is very plausible but isolation and structure demonstration using extensive spectral techniques (NMR, CD …) will be necessary to verify this assumption.

The next increased compound in fMT, with a fold at 1.6, is eluted at Rt 8.44 min and shows a negative molecular ion at *m*/*z* 197.0823 [M-H]^−^ that is compatible with the molecular formula C_10_H_14_O_4_. This 10-carbon compound points to a thymol-type structure, as described above. Its negative ion MS/MS spectrum, depicted in Figure 4, is fully compatible with a trihydroxythymol.

Despite the possibility of several isomers, we propose the putative 8,9,10-trihydroxythymol for this annotation, owing to the fact that this derivative has been isolated in various genera of the Asteraceae family [23,38,39].

All the proposed structures of the annotated compounds of the fMT are summarized in Figure 5.

The dereplication of peaks that increased in the mother tinctures prepared from dried whole *A. montana* (dMT) plants led to the annotation of nine compounds (Table 1 and Figure 6). Following the chronology of retention times, the first peak at Rt 1.09 shows a positive molecular ion at *m*/*z* 116.0704, suitable for the molecular formula C_5_H_9_O_2_N and referring, a priori, to the structure of the amino acid proline, which is frequently encountered in plant extracts. However, the MS/MS product ions (Appendix A) of the parent *m*/*z* 116 were not consistent with this deduction, due to a significant neutral loss of 32 Da, leading to the *m*/*z* 84 base peak. A putative methoxylated pyrrolidinone structure is proposed for this compound on the basis of the MS/MS fragments (Appendix A), supported by the literature, wherein a pyrrolidine derivative has previously been identified in *A. montana* [26].

The following three components listed in Table 1 present even pseudo-molecular ions in either positive or negative ion modes, this being indicative of nitrogenous compounds. Surprisingly, they appear to be particularly overexpressed in dMT, since their folds range from 104 to 577. For the Rt 1.23 min peak, it is calculated that the molecular formula C_11_H_21_O_7_N refers to only 2 occurrences in the DNP database [35], one of them being fructose-valine (Figure 6). Close examination of the MS/MS fragmentation of its [M-H]^−^ parent is fully compatible with the proposed structure, as depicted in Figure 7. Indeed, in the ESI-MS/MS negative ion mode, this compound follows a very classical fragmentation pattern, as described for natural heterosides and particularly for the flavonoid C-glycosides, in which [M-H-90]^−^ and/or [M-H-120]^−^ are characteristic product ions formed by cross-ring cleavages in a sugar residue and are present in great abundance [40]. According to these considerations, the amino acid nature of this sugar conjugate compound is verified, attested to by an *m*/*z* 116 ion that is consistent with a valine moiety, but the structure of the sugar remains to be clarified. It is worth noting that sugar amino acid conjugates are well known in food chemistry since they are important intermediates involved in an early Maillard reaction [41]. The Maillard reaction is a common term for a broad array of reactions that typically involve reducing sugars, such as glucose, and aliphatic amino groups of biological molecules. A sophisticated reaction cascade is initiated by the formation of a Schiff base (a glycosylamine) between a reducing sugar and an amino acid and/or the amino acids of peptides or proteins. In the presence of a nucleophilic catalyst, glycosylamine derived from an aldose can rearrange into a more stable 1-amino-1-deoxy-2-ketose; such a reaction is called an Amadori rearrangement, leading to an Amadori rearrangement product (ARP) that is dependent on the amino acid involved in the reaction. When glycosylamine is derived from a ketose sugar, it can rearrange into a 2-amino-2-deoxyaldose, which is also termed a Heyns rearrangement product (HRP). The successive breakdown of ARPs or HRPs generates volatile carbocyclic and heterocyclic compounds, oligomers and polymers, thus bringing various flavors and yellowish to brown colors to food products [27,32]. Concerning the dereplication of these rearrangement products, it should be noted that for the same amino acid involved in the Maillard reaction, the rearrangement products (i.e., the Schiff base, ARP, and HRP) are isomers, as depicted in Appendix A. Therefore, it must be determined which type of product is generated in the mother tincture of arnica prepared from the dried whole plant.

Fortunately, the positive ionization electrospray MS/MS of amino acid glycoconjugates has extensively been studied [27,28,42]. This means that the diagnostic product ions derived from the [M+H]^+^ parent can be used for discriminating between the three types of products. Combining the data published in the literature, it appeared that the [Amino Acid-H+CH_2_]^+^ ion, although weak (less than 10% relative intensity) is characteristic of Amadori compounds. The presence of this ion in the MS/MS spectrum of the putative valine glycoconjugate positive protonated ion at *m*/*z* 280 (Appendix A) is de facto in favor of a fructose (i.e., Amadori) valine conjugate. This fructose moiety is shown in Figure 6 in its β-pyranose form since it was identified as predominating in solution and at pH 7, but acyclic and α- and/or β-furanose structures are present in tautomeric equilibria and in various proportions, depending on the pH [35,36,37]. Following these findings, a second Amadori rearrangement product, fructose leucine or isoleucine (the two isomers are both possible) was annotated at Tr 1.35 min, due to its MS^2^ fragments being identical to those of fructose valine but with a 14 Da shift (Table 1 and Appendix A). As well as valine-fructose, (iso)leucine-fructose is a frequently encountered ARP in plants [30,31,32]. In contrast, a third derivative, annotated as an ARP and increased by more than 570 times in dMT, has never been described previously. Its peak at Rt 1.25 min possesses a molecular formula of C_11_H_17_O_8_N and a negative ion MS/MS fragmentation profile that is very close to that of leucine-fructose (see Appendix A). Its positive-ion MS^2^ spectrum obtained from the [M+H-H_2_O]^+^ parent at *m*/*z* 274 (Table 1) presents the [amino acid-H+CH_2_]^+^ diagnostic ion at *m*/*z* 142, thus suggesting an ARP derivative. Therefore, it can be deduced that the molecular formula is C_5_H_7_NO_3_ for the corresponding amino acid. Only two occurrences appear with an amino acid structure in the DNP database [35] for this formula; these correspond to two oxidated proline derivatives or isomers (oxoprolines or hydroxydehydroproline). We propose a 4-oxoproline structure (Figure 6) for this putative ARP, but isolation and full structure identification using extensive spectroscopic techniques will be necessary to fully identify this new Amadori rearrangement product.

The last five ions enhanced in the dMT were easy to annotate since they have previously been identified in *A. montana* flowers [16,17]. They were all tentatively assigned as dicaffeoylquinic acid derivatives (Figure 6), substituted by one (the four regioisomers at *m*/*z* 601, [M-H]^−^) or two methoxyoxaloyl moieties (*m*/*z* 687, [M-H]^−^). The latter compound has no isomers since the extracted ion chromatogram of the *m*/*z* 687 shows only one peak (Appendix A). This produced the MS/MS base peak at *m*/*z* 601 (Table 1) due to the loss of a methoxyoxaloyl residue (−86), which suggested that one of the two methoxyoxalic acid residues is found at the C-1 position of the quinic acid moiety, as proposed by Jaiswal and Khunert, who have thoroughly investigated the quinic acid derivatives present in *A. montana* by negative ion ESI MS^n^ [17]. Based on their data, we propose the same 3,5-dicaffeoyl-1,4-dimethoxyoxaloyl quinic acid isomer here, but it should be stressed that this compound has only been identified by LC-MS and that Lin and Harnly proposed another regioisomer (a 1,5-dicaffeoyl-3,4-dimethoxyoxaloyl derivative) [16]. It is also important to note that, by investigating the MS/MS spectrum of the *m*/*z* 687 ion, we observed (as shown in Table 1 and Appendix A) unusual fragments linked to the presence of the methoxyoxaloyl moieties. This concerns the loss of two successive CO_2_ [*m*/*z* 643 (−44) and 599 (−2 × 44)] from the pseudomolecular ion *m*/*z* 687. Those losses explain the relatively abundant diagnostic ions at *m*/*z* 437 and 275, which have been interpreted as the [M-H-2CO_2_-caffeoyl]^−^ and [M-H-2CO_2_-2caffeoyls]^−^ ions, respectively. We suggest that this decarboxylation occurs as a methoxyoxaloyl residue involving a rearrangement leading to a new acetyl residue, as shown for the [M-H-2CO_2_-caffeoyl]^−^ fragment in Appendix A. To our knowledge, this is the first time that this kind of “substitution” of an ester by another is described in MS^2^ fragmentations; further analysis (for example by MS^n^) would be interesting to confirm or refute this finding. The four regioisomers at *m*/*z* 601 were correctly separated (~25–30 min), as illustrated in the extracted ion chromatogram (Appendix A). These isomers were putatively assigned as dicaffeoyl-methoxyoxaloyl quinic acids (DCMOQA), according to their characteristic MS/MS product ions (Table 1 and Appendix A). The four compounds lose a methoxyoxaloyl moiety leading to *m*/*z* 515 with different relative abundances and present the same base peak at *m*/*z* 395 (−44–162 = −162), corresponding to the neutral losses of CO_2_ and caffeoyl molecules. According to Jaiswal and Kuhnert [17], it is possible to distinguish the four regioismers since the four corresponding MS/MS produced some secondary peaks with different relative abundances, as observed above for the *m*/*z* 515 ions. From these findings and according to the works of Jaiswal and Kuhnert [17] and before Clifford and co-workers [34], a high *m*/*z* 515 fragment coupled to a high *m*/*z* 439 is indicative of 1,5-DC-3-MOQA, a high *m*/*z* 515 and a low *m*/*z* 439 value points to 1-MO-4,5-DCQA, and low *m*/*z* 515 were observed for 1,3-DC-4-MOQA and 3,5-DC-4-MOQA. The distinction between the two latter regioisomers can be achieved by the relative abundances of the *m*/*z* 439 ions, which is more important for the 3,5-DC-4-MOQA when compared to 1,3-DC-4-MOQA [17]. These deductions were consistent with the results obtained by Lin and Hanrly [16] in terms of the order of elution of these derivatives by reversed-phase chromatography and their abundance in *A. montana,* where the 1,3-DC-4-MOQA has the highest concentration (see Appendix A).

## 3. Discussion

This study examined the differences in the phytochemical compositions of homeopathic mother tinctures prepared from the fresh or dried whole plants of *A. montana*, using an LC-MS-based multiplexed metabolomic approach. The multivariate statistical analysis revealed significant differences between “fresh” and “dry” mother tinctures and permitted us to highlight the more obvious distinguishing features. It is notable that the sesquiterpene lactones were not affected by the drying process. This is an important result since they are considered to be the main metabolites responsible for the bioactivities of arnica [4]. The same can be said concerning flavonoids. This finding is contrary to the results obtained by Asadi and colleagues, who noted an increase in the flavonoid content of the flowerheads of *Arnica chamissonis* after oven-drying at 40 °C [44]. In order to explore the reasons for this difference in results, a multivariate analysis (S-plot) targeting flavonoids is shown in Appendix A. It shows that flavonoids are slightly more concentrated in the dMT but were not detected as discriminating compounds with the set treshold parameters. The annotation, with a human mind (*in silico* annotations were particularly disappointing (data not shown)), of discriminant metabolites showed that caffeoyl quinic acid esters and the newly described compounds in *A. montana*, such as hydroxysphingenine and thymol derivatives, were increased in MTs prepared from the fresh whole plant, whereas methoxyoxaloyl dicaffeoyl quinic acid esters (MODCQA) and, particularly, Amadori rearrangement products were increased during the drying process.

We are unable to proffer an explanation concerning the increase of MODCQA in dMTs, but the formation of amino acid glycoconjugates during drying or cooking was thoroughly explored in the context of foods some time ago [41]. Their occurrence in herbal drugs is much less widely documented. However, it can be stated the roots of ginseng (*Panax ginseng* C. A. Meyer), which is traditionally processed to make white ginseng (with the roots air-dried after peeling) and red ginseng (with the roots steamed at 98–100 °C without peeling). It is considered that steaming enhances the biological activities of red ginseng and that this enhancement can be related to ARPs [45] that were shown to exhibit antioxidant activities [46,47]. Other bioactivities have been reported for Maillard reaction products (MRP) in general (i.e., from early intermediates, such as ARP or HRP, or glycated amino groups of peptides/proteins, to final products such as melanoidins or volatile derivatives, both from vegetal and animal origin). Notable qualities include antibacterial activities [48], anti-inflammatory effects [49], bifidogenic properties [50], and angiotensin-converting enzyme-inhibitory activities [51]. On the other hand, several negative aspects of MRP cannot be ignored. For example, non-enzymatic browning (final products of the Maillard reaction) is a limiting factor that can shorten the shelf life of some food products, such as human milk [52]. In humans, the Maillard reaction also occurs between the free amino groups of proteins and reducing sugars, leading to the production of so-called advanced glycation end products (AGE) that have been linked to the increased prevalence of several diseases, such as diabetes, atherosclerosis or neurodegenerative diseases [53,54,55]. Therefore, both beneficial and adverse aspects of MRP seem to coexist (for a review, see [56]) and further research on systematic stability and toxicology, linked to ARP and MRP, are needed. Thus, it seems important to control the processing conditions and limit the progression of the Maillard reaction to reach these advanced stages. The detection of early-stage products, such as ARP, by negative and/or positive ion ESI-MS/MS, as described in this work, can contribute to monitoring the progression of the Maillard reaction and increasing the quality control and standardization of herbal medicinal products in general and those containing *A. montana* in particular.

Another type of metabolite of interest that was highlighted by multivariate statistical analysis concerned the thymol derivatives. Two compounds that increased in fMT were tentatively identified: trihydroxylthymol, newly described in *A. montana,* and a new putative natural product, 8,9-epoxy-10-hydroxy-thymolisobutyrate. Thymol, its derivatives, and the extracts containing them are well known and used, among others, for their antibacterial and antibiofilm properties [57]. These are typically the metabolites that are encountered in essential oils (particularly in *Thymus* spp.) and, for *A. montana*, it was reported by Weremczuk-Jezyna and coworkers that these compounds are located in the essential oils obtained from roots and hairy roots [37]. Interestingly, the fact that thymol-type compounds are present in the MT of *A. montana* may indirectly contribute to the healing properties of arnica since they contain topical medications due to their antiseptic and antiphlogistic properties [58]. This explains research interest in the use of the MT of *A. montana* for topical preparations, since these are officially (according to the European and French Pharmacopeia [10,11]) prepared from the whole fresh plant, so this includes the roots. It will, however, be interesting to investigate the thymol derivative content of the flowerheads of arnica to confirm this assumption, but the yield of the essential oil obtained from flowers is about twenty times lower than that of the roots [57]. If the plants are lacking in flowerheads, thymol derivatives may constitute interesting quality markers by which to distinguish the two possible raw materials to produce tinctures, as used for phytopharmaceuticals (dried flowerheads) or homeopathy (the fresh whole plant).

A final compound that is newly described in *A. montana* and enhanced in fMT is the putative 4-hydroxy-8-sphingenine, commonly called dehydrophytosphingosine. Sperling and Heinz reviewed the functions of sphingolipids in plants and reported their importance in cell signaling, membrane stability, the abiotic stress response, programmed cell death, and plant-pathogen interactions [21]. The biological properties of phytosphingosine, the saturate analog of dehydrophytosphingosine, are well documented since phytosphingosine is a cosmetic ingredient and has several potential applications, particularly in dermatology. For example, phytosphingosine might serve as an effective melanogenesis inhibitor in melanocytes and could be used as a skin-whitening product [59]. Phytosphingosine is also found in the stratum corneum of the skin, contributing to the skin barrier function and displaying anti-inflammatory and anti-microbial activities [60,61,62]. An acetylated derivative of phytosphingosine was shown to exert an inhibitory action on angiogenesis through the inhibition of mitogen-activated protein kinase activation and intracellular calcium increase, thereby affecting the process of wound healing [63]. The phosphorylation of phytosphingosine also exerts an influence on biological activities in in vitro human dermal fibroblasts by promoting the activity of epidermal growth factor and attenuated H_2_O_2_-induced cell growth arrest [64,65]. Thus, dehydrophytosphingosine is another newly annotated compound in arnica that could contribute to the activities of arnica on damaged skin.

## 4. Materials and Methods

### 4.1. Sample Preparation

The whole flowering *A. montana* plants (320–786 kg, depending on the supplier) were acquired from six different French suppliers (EARL du Patuet, Jourd’hui, Herboristerie de la Chartreuse, Horizon Nature, Sicarappam, SICA Vivaplantes). The herbs were manually harvested during the summer of 2020 from five different geographic areas (4 areas in France: Massif du Morvan, Massif Central (2 samples), Ardèche, the Auvergne, and 1 area in Germany: Hesse) and were checked (macroscopic and microscopic botanical identification) by Boiron Laboratories (Messimy, France) and recorded under vouchers nos. 20066019, 20066025, 20066052, 20066110, 20076001, and 20066041 for EARL du Patuet, Jourd’hui, Herboristerie de la Chartreuse, Horizon Nature, Sicarappam, and SICA Vivaplantes, respectively. It is worth noting that *A. montana* is included in the International Union for the Conservation of Nature (IUCN) list of endangered species [66] and harvesting is regulated and confined to several areas in France and Germany.

The mother tinctures were produced by Boiron Laboratories. For this purpose, each independent batch of *A. montana* plants from the different geographic areas was divided into two parts. One part was used as the fresh raw material, according to the European Pharmacopoeia monograph 2371 (Methods of preparation of homeophatic stocks and potentization), method 1.1.10 [10], and the other part was dried by the firm “Herboristerie de la Chartreuse et du Grésivaudan” (France) in a multi-layer dryer (30 m^3^ AMB-Rousset type) with circulating airflow for 2 h at 65 °C. The manufacturing steps of the 1.1.10 method prescribed in monograph 2371 were followed. The fresh and dried plants were macerated in duplicate for 21 days in a final 45% ethanol (*v*/*v*) solution (DER 1:10) and then filtered to obtain the mother tinctures. These were aliquoted and stored et +4 °C in opaque 30 mL vials in duplicate before use. This process allowed us to obtain the tincture-related batches from fresh and dried plants. Overall, 24 samples were analyzed (12 fMT (6 different origins extracted in duplicate) and 12 dMT) with one pool of each tincture in triplicate (QC samples). LC-MS chromatograms of dried and fresh samples and depicting the positions of the discriminant compounds can be viewed in the Appendix A.

### 4.2. Determination of Dry Residues

Five mL of the mother tincture were weighed in a dried and pre-weighed vial, then the samples were dried under the flow of nitrogen for 24 h. The dry residues were determined in duplicate for each sample, and the result is directly expressed as a percentage (*m*/*v*) of the material.

### 4.3. Ultra-High-Performance Liquid Chromatography-Orbitrap Analysis

Mother tinctures were separated using a DionexTM Ultimate 3000 UHPLC system, which includes an Ultimate 3000 RS pump, an Ultimate 3000 RS autosampler, and an Ultimate 3000 column compartment. The analytes were detected using an Ultimate 3000 DAD for UV detection (200 to 400 nm) and LTQ Orbitrap XL for mass detection (Thermo Scientific^TM^, Hemel Hempstead, UK). The mass detection was performed using an electrospray ionization (ESI) source in positive (PI) and negative ionization (NI) modes at 15,000 resolving power (full width at half-maximum (FWHM)) at *m*/*z* 400). The mass scanning range was *m*/*z* 100–2000 Da, the capillary temperature was 300 °C, and the ionization spray voltage was 3.5 kV. Each full scan was followed by data-dependent MS/MS on the four most intense peaks using stepped collision-induced dissociation (CID) at 35 arbitrary units (isolation width 1 *m*/*z*, activation Q 0.25).

Various assays were carried out in order to find the best conditions for separation and MS detection of the highest number of compounds. Thus, the separation of the most polar compounds using a Hydrophilic Interactions Liquid Chromatography (HILIC) column did not give satisfactory results due to very broad peaks. Similarly, reverse phase tests using acidified acetonitrile/water mixtures did not give good separations. For MS detection, the automatic tune function of the spectrometer was used to optimize parameters by selecting the *m*/*z* 515 ion (quinic acid derivative) in negative ion mode and the *m*/*z* 333 ion (sesquiterpene lactone ester) in positive ion mode. Therefore, gra-dient elution was performed on a C18 Acquity column (150 mm, 2.1 mm i.d, 1.7 µm, Waters, MA, USA) with a guard column at 40 °C in the column compartment. A gradient elution of solvent A (0.1% formic acid in water) and solvent B (methanol) was applied as follows: 0–3 min, 10% B; 3–9 min, 10–15% B; 9–30 min, 15–35%; 30–45 min 35–60% B; 45–50 min, 60–80% B; 50–55 min, 80% B; 55–58 min, 80–10% B. The injected volume sample was set at 3 µL and the flow rate for elution was fixed at 0.4 mL/min.

### 4.4. Data Processing

Raw data from UHPLC-HRMS in the positive and negative ionization modes were converted into an .abf file (Reifycs Abf Converter, Tokyo, Japan) and processed with MS-DIAL 4.48 [67]. Signal was extracted at between 50 and 2000 Da and between 0 and 60 min. Mass variation in the centroid mode was set at 0.01 Da for MS^1^ and at 0.025 Da for MS^2^. The peaks were aligned on quality control (QC), with a retention time tolerance of 0.05 min and a mass tolerance of 0.015 Da. Two detection thresholds were used, with a mass slice width of 0.1 Da, the first at 7 × 10^4^, allowing us to recover 924 peaks, and the second at 1.2 × 10^7^, allowing us to recover 71 peaks. The adducts and complexes were identified to exclude them from the final peaks list.

The peaks lists have been concatenated and cleaned using the MS-CleanR program on the shiny interface of R studio 4.0.3 [68]. The minimum blank ratio was set at 0.8, with a maximum mass difference of 0.01 Da and a maximum retention time tolerance of 0.025 min. A Pearson correlation of 0.8 (minimum) was used for clusterization, with a maximum *p*-value of 0.05. The final peak list is directly exported in comma-separated value (CSV) format, prior to multivariate data analysis.

### 4.5. Statistical Analysis

For multivariate data analysis, the CSV files were directly imported into SIMCA^®®^ 14.0 (Sartorius Stedim Biotech, Umea, Sweden). Data were scaled to unit variance for principal component analysis (PCA). The dataset was centered and Pareto-scaled to build a supervised model, using orthogonal projection for latent structure-discriminant analysis (OPLS-DA), with the fresh (F) and dried (D) classes as the Y inputs. The quality of the OPLS-DA model was evaluated by the goodness-of-fit parameter (R^2^) and the predictive ability parameter (Q^2^).

For dried residues, Student’s t-test was used for statistical comparisons. The *p*-values of <0.05 were considered to be statistically significant.

The fold value represents the enrichment in metabolites in one extract relative to the other. This value was determined for each feature by a ratio between the average area (MS base peak chromatograms) in the dried MT and fresh MT. Results here are always expressed superior to 1, thus representing an enrichment of the metabolites in the selected MT.

## Figures and Tables

**Figure 1 molecules-27-02737-f001:**
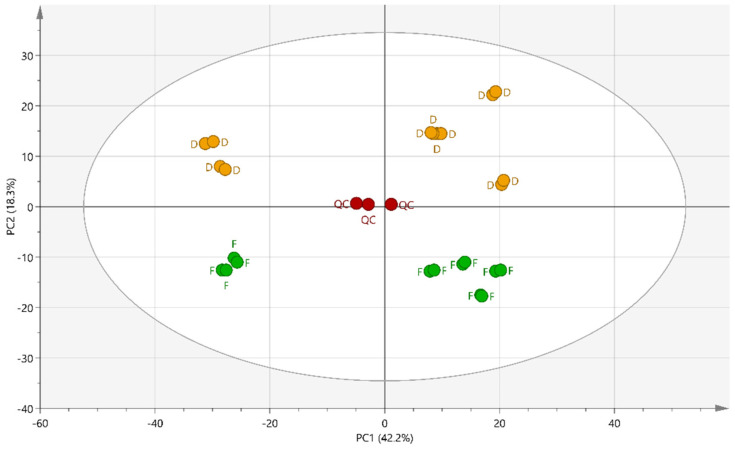
PCA score plot of the first two principal components (PC1 and PC2), based on the LC-MS dataset, including 924 compounds. Mother tinctures from the fresh plants (F), mother tinctures from dried plants (D) and quality control (QC).

**Figure 2 molecules-27-02737-f002:**
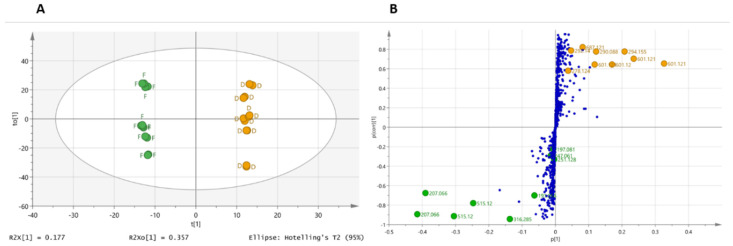
Orthogonal partial least squares discriminant analysis (OPLS-DA) between the mother tinctures prepared from fresh (F) and dried (D) whole plants of *Arnica montana* in positive- and negative-ion ESI-MS. (**A**) OPLS-DA score plot, (**B**) OPLS-DA loading S-plot.

**Figure 3 molecules-27-02737-f003:**
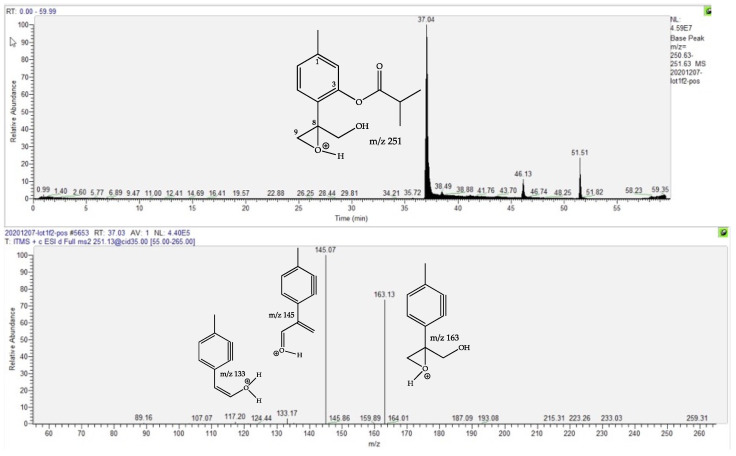
Ionic current chromatogram of the *m*/*z* 251.15 positive ion (fMT) corresponding to the proposed 10-hydroxy-8,9-epoxy-thymolisobutyrate (**above**) and the MS/MS spectrum of the *m*/*z* 251.15 ion, with proposed structures for fragment ions (**below**). The position of the positive charge is arbitrary when several oxygens are present.

**Figure 4 molecules-27-02737-f004:**
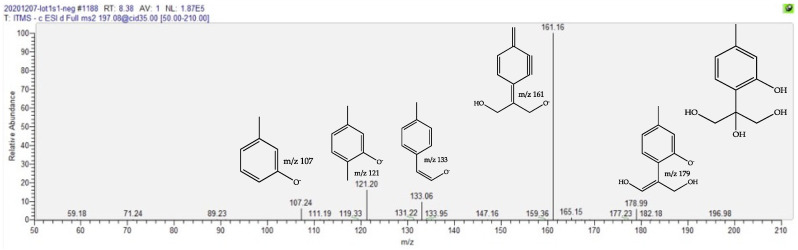
Negative-ion MS/MS spectrum of the [M-H]^−^ parent trihydroxy thymol at *m*/*z* 197, with proposed structures for the product ions.

**Figure 5 molecules-27-02737-f005:**
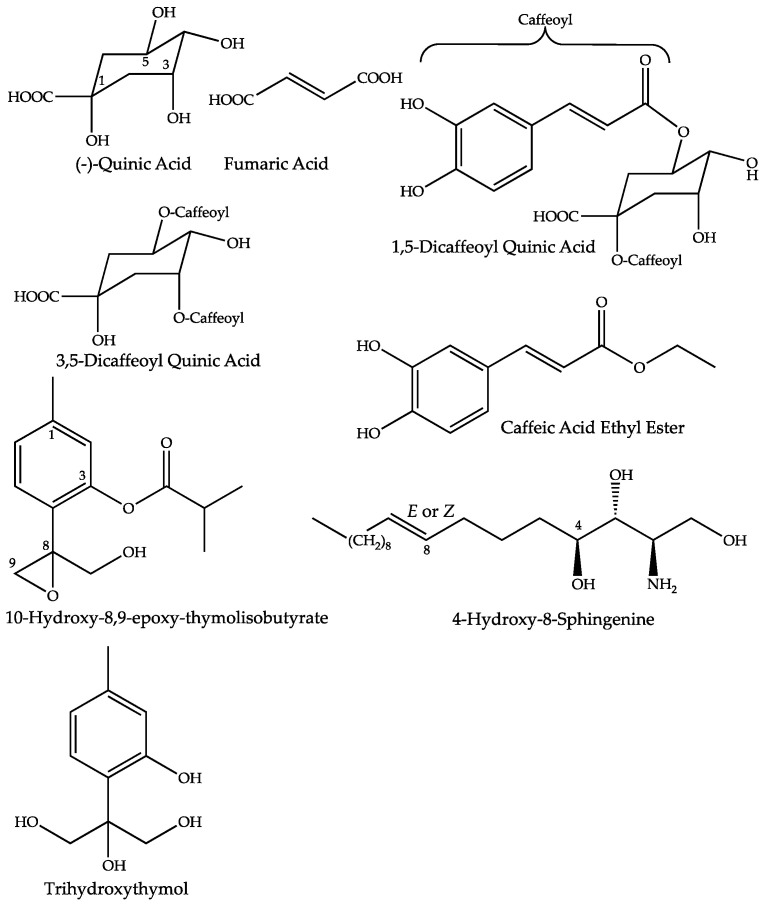
Proposed structures for compounds increased in the mother tinctures, prepared from fresh whole plants of *Arnica montana.*

**Figure 6 molecules-27-02737-f006:**
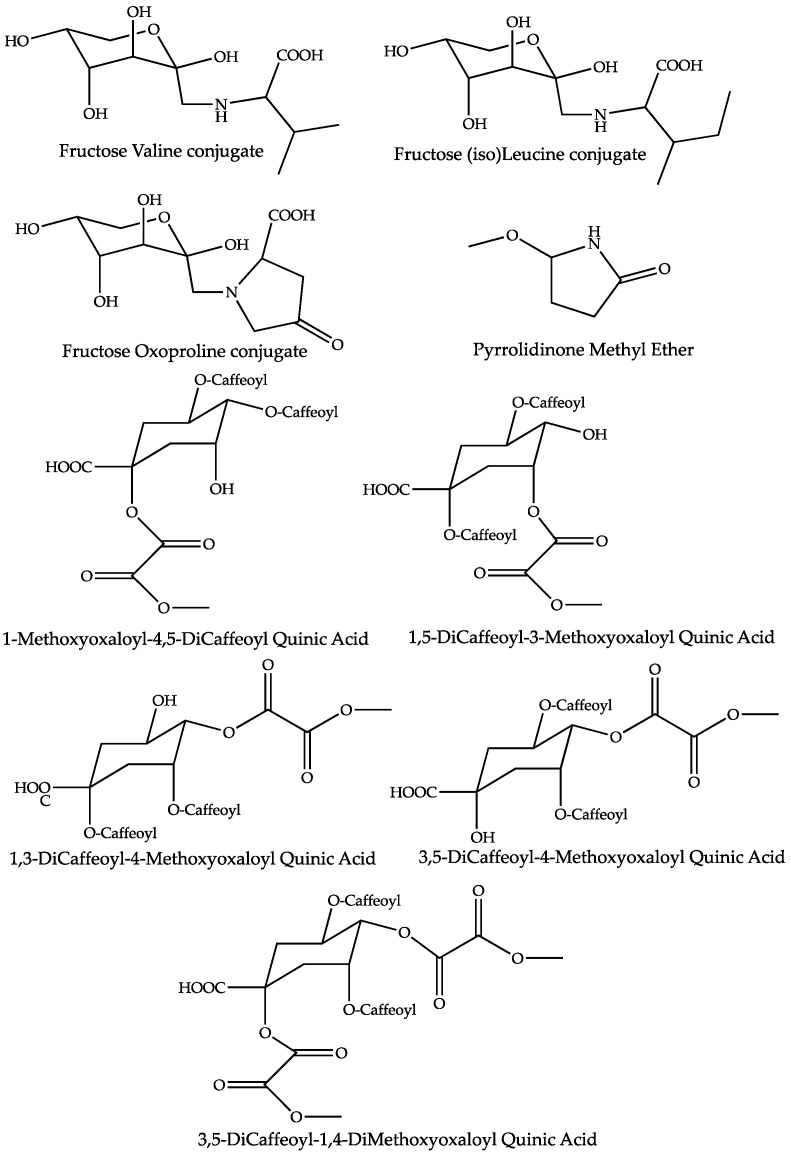
Proposed structures for the compounds increased in mother tinctures prepared from the dried whole plant of *Arnica montana*.

**Figure 7 molecules-27-02737-f007:**
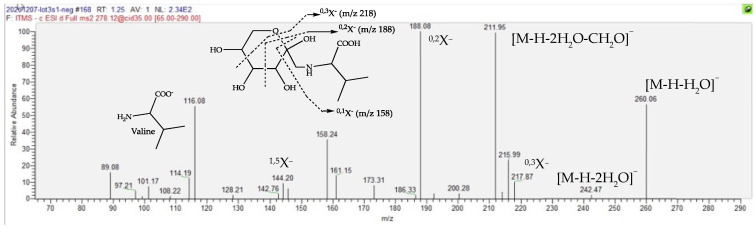
MS/MS spectrum of the [M-H]^−^ ion at *m*/*z* 278, showing the structures of the diagnostic product ions, mainly due to the characteristic cleavages occurring in the hexose moiety. These product ions from glycoconjugates are denoted according to the nomenclature introduced by Domon and Costello [43].

**Table 1 molecules-27-02737-t001:** List with MS and MS/MS data of annotated compounds increased in each type of mother tinctures. RI: Relative Intensity in %, bp: occurs as base peak.

Proposed Identification	Fold	Rt (min)	Molecular Formula	Experimental [M-H]^−^ (Error in ppm)	Negative ion MS/MS fragments (RI)	Experimental [M+H]^+^ (Error in ppm)	Positive Ion MS/MS Fragments (RI)	References
Compounds increased in MT prepared from fresh plant								
Quinic Acid	2.2	1.06	C_7_H_12_O_6_	191.0562 (0.36)	173 (bp), 127, 111, 93, 85			[7,16,17]
Fumaric Acid	1.4	1.33	C_4_H_4_O_4_	115.0039 (2.25)	97 (bp)			[7,16,17]
1,5-Dicaffeoyl-Quinic Acid	1.4	25.63	C_25_H_24_O_12_	515.1201 (0.60)	353 (bp), 335 (10), 191 (18)			[7,16,17]
3,5-Dicaffeoyl-Quinic Acid	1.5	25.69	C_25_H_24_O_12_	515.1201 (0.60)	353 (bp), 335 (10), 203 (1) 191 (18), 179 (4), 173 (1)			[7,16,17]
Caffeic Acid Ethyl Ester	2.2	31.90	C_11_H_12_O_4_	207.0659 (1.70)	179 (bp), 135 (20)			[18]
4-Hydroxy-8-Sphingenine	3.1	50.89	C_18_H_37_O_3_N	None		316.2849 (0.13)	298 (bp), 280 (75), 262 (5)	[19,20,21]
10-Hydroxy-8,9-epoxy-thymolisobutyrate	1.3	37.07	C_14_H_18_O_4_	249.1136 (1.59)		251.1279 (0.75)	163 (75), 145 (bp)	[22,23,24,25]
Trihydroxy Thymol	1.6	8.44	C_10_H_14_O_4_	197.0823 (1.41)	179 (10), 161 (bp), 133 (10), 121 (15)			[22,23,24,25]
Compounds increased in MT prepared from dried plant								
Methoxy Pyrrolidinone	2.2	1.09	C_5_H_9_O_2_N	None		116.0704 (0.90)	98 (22), 88 (18),84 (bp), 56 (18)	[26]
Fructose Valine conjugate	151.1	1.23	C_11_H_21_O_7_N	278.1245 (0.05)	260 (20), 218 (20), 212 (40), 188 (bp), 158 (15, 116 (30))	280.1391 (0.08)	262 (bp), 244 (2), 216 (2), 130 (2)	[27,28,29,30]
Fructose Oxoproline conjugate	577.2	1.25	C_11_H_17_O_8_N	290.0880 (1.55)	272 (20), 254 (10), 200 (bp), 170 (10), 128 (10)	292.1026 (0.32)	* 256 (90), 238 (bp), 142 (2), 130 (40)	[31]
Fructose (Iso)Leucine conjugate	103.8	1.35	C_12_H_23_O_7_N	292.1400 (0.33)	274 (20), 226 (55), 202 (bp), 172 (15), 130 (30)	294.1548 (0.03)	276 (bp), 258 (4), 230 (4), 146 (4), 144 (2)	[30,31,32]
1-Methoxyoxaloyl-4,5-DiCaffeoyl Quinic Acid	1.5	25.45	C_28_H_26_O_15_	601.1200 (0.18)	557 (20), 515 (80), 439 (15), 395 (bp), 377 (18), 299 (10), 233 (5)			[16,17]
1,5-DiCaffeoyl-3-Methoxyoxaloyl Quinic Acid	1.2	25.95	C_28_H_26_O_15_	601.1203 (0.68)	515 (90), 439 (70), 395 (bp), 377 (15), 233 (30), 173 (5)			[16,17]
3,5-DiCaffeoyl-4-Methoxyoxaloyl Quinic Acid	1.5	28.54	C_28_H_26_O_15_	601.1203 (0.68)	515 (40), 439 (38), 395 (bp), 377 (10), 233 (28), 173 (5)			[16,17]
1,3-DiCaffeoyl-4-Methoxyoxaloyl Quinic Acid	1.4	30.60	C_28_H_26_O_15_	601.1195 (0.48)	515 (10), 439 (15), 395 (bp), 233 (5)			[16,17]
3,5-DiCaffeoyl-1,4-DiMethoxyoxaloyl Quinic Acid	8.9	29.51	C_31_H_28_O_18_	687.1208 (0.76)	601 (bp), 599 (25), 557 (30), 437 (40), 275 (25)			[16,17]

* MS/MS of *m*/*z* 274 ion {M-H_2_O+H}^+^.

## Data Availability

Not applicable.

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
