# Peer review of "Comparison of the Phytochemical Variation of Non-Volatile Metabolites within Mother Tinctures of Arnica montana Prepared from Fresh and Dried Whole Plant Using UHPLC-HRMS Fingerprinting and Chemometric Analysis"

_molecules, 2022, doi:10.3390/molecules27092737_

Round 1

Reviewer 1 Report

The manuscript described an untargeted metabolomic approach that using UHPLC-HRMS/MS to compare the phytochemical composition of dried and fresh mother tinctures of A. montana. The results were analyzed using several chemometrics approach such as PCA and oPLS-DA. The distinguished metabolites from the dried and fresh mother tinctures of A. montana were identified based on the MS fragmentation and literature. The results were clear and quite solid.

  1. The Figure S1 plot could be changed into a boxplot with dots in which each individual sample amount was addressed to facilitate the readers.
  2. For Figure 1, a scree plot of the PCA model could be addressed into the supplementary information. And how about a cluster circle of the two groups to check the potential outlier of the separation?
  3. The literature mentioned in the Section 2 for the compound structure identification could be added into Table 1. The percentage unit of RI could be addressed either in the Table 1 title. And what is the cutoff of the fold change in Table 1? Did the author do a fold change cutoff before construct the oPLS-DA model or not? And if not, why a compound such as 1,5-DiCaffeoyl-3-Methoxyox-aloyl Quinic Acid that only represents 1.2-fold difference could contribute significantly to the oPLS-DA model?
  4. Two representative chromatograms of dried and fresh sample with both positive and negative mode could be added separately into the manuscript and/or the supplementary information. If possible, the distinguished peak (metabolites) should be addressed on the representative chromatograms.
  5. It is very interesting to show the annotation chromatograms and/or tables in the supplementary information about the flavonoids with structural information. Because flavonoids can be increased in a certain temperature and then decrease when the temperature went even higher [1]. Therefore, the results difference might cause by the temperature changes. While this also depends on the fold change cutoff of the model and how much the flavonoids decreased to become non-significant to the model.

Reviewer 2 Report

If the title of the publication and its topic is UHPLC-HRMS fingerprinting and chemometric analysis, then the entire introduction should be a lietrature review of both (UHPLC HRMS and chemometrics) This is not the case. Please change the entire introduction and do a thorough, critical literature review.

line 86-87: the authors write about over 900 compounds. I would like to see them identified.

When reading this publication, it is difficult to know what is new about this work. I wish the authors had made this clear. It seems to me that the novelty of this research is limited. Both untargeted metabolomics, UHPLC HRMS, and chemometrics have been used many times in nalogical studies. So why are these unique.

Please complete this publication with comprehensive information on how chromatographic conditions (column, mobile phase composition, gradient) and detection conditions (all MS parameters) were chosen.

lines 362-365: if such complex studies result in such trivial conclusions (which can be made without doing the studies), I don't understand their point. Without doing UHPLC HRMS analysis I know that differences between "fresh" and "dry" mother tinctures will be large.

Round 2

Reviewer 2 Report

The authors have responded sensitively to all the reviewer's comments. The publication has gained in value and can be published.